# Synthesis of Cyclic Peptides in SPPS with Npb-OH Photolabile Protecting Group

**DOI:** 10.3390/molecules27072231

**Published:** 2022-03-29

**Authors:** Tingting Chen, Gang Wang, Lin Tang, Hongpeng Yang, Jing Xu, Xiaoxue Wen, Yunbo Sun, Shuchen Liu, Tao Peng, Shouguo Zhang, Lin Wang

**Affiliations:** Institute of Radiation Medicine, Beijing 100850, China; ctt1213340553@126.com (T.C.); adamtl@126.com (L.T.); 18211168112@163.com (H.Y.); luciferxjj@126.com (J.X.); crystalcat@vip.sina.com (X.W.); sunyunbo0919@126.com (Y.S.); liusc118@sohu.com (S.L.)

**Keywords:** photolabile protecting groups, Npb-OH, cyclic peptides

## Abstract

Significant efforts have been made in recent years to identify more environmentally benign and safe alternatives to side-chain protection and deprotection in solid-phase peptide synthesis (SPPS). Several protecting groups have been endorsed as suitable candidates, but finding a greener protecting group in SPPS has been challenging. Here, based on the 2-(*o*-nitrophenyl) propan-1-ol (Npp-OH) photolabile protecting group, a structural modification was carried out to synthesize a series of derivatives. Through experimental verification, we found that 3-(*o*-Nitrophenyl) butan-2-ol (Npb-OH) had a high photo-release rate, high tolerance to the key conditions of Fmoc-SPPS (20% piperidine DMF alkaline solution, and pure TFA acidic solution), and applicability as a carboxyl-protective group in aliphatic and aromatic carboxyl groups. Finally, Npb-OH was successfully applied to the synthesis of head–tail cyclic peptides and side-chain–tail cyclic peptides. Moreover, we found that Npb-OH could effectively resist diketopiperazines (DKP). The α-H of Npb-OH was found to be necessary for its photosensitivity in comparison to 3-(*o*-Nitrophenyl)but-3-en-2-ol (Npbe-OH) during photolysis-rate verification.

## 1. Introduction

With advances in biotechnology, many active peptides and proteins have been developed into drugs, been used in clinical practice, and played vital roles in maintaining the normal functions of the body due to their high specificity, low required dosages, clear functions, and low numbers of side effects [1]. However, linear peptides are unstable in the body, susceptible to degradation, and easily removed by the kidneys, leading to short half-lives. Therefore, the study of long-acting peptide drugs has become a hot topic. Studies have shown that a polypeptide’s cyclization can stabilize its dominant configuration, effectively improving its stability against proteases and, thereby, its pharmacokinetic properties and biological activity [1]. The self-cyclization of the amino acid side chain on a polypeptide includes head-to-tail, head-to-side, and side-to-tail processes [2]. Traditional synthetic cyclic peptide methods rely on three dimensions of orthogonal protecting groups to allow for the synthesis of the linear peptide, the selective deprotection of the reactive ends (N- or C-terminus or side chain), cyclization, and then the final cleavage from the solid support. A common third-dimensional protecting group is an allyl group, which is easy to synthesize, has high acid–base tolerance, and can be catalytically removed with Pd reagents. However, Pd reagents are expensive, and the endpoint of the reaction is difficult to monitor, limiting their further application. A recent review of the greening of photolabile protecting groups (PPGs) showed that the allyl group has advantages for the compound protection and deprotection processes in lowering the environmental burden. The greening of the SPPS process has also seen improvements in chain assembly, resin cleavage, side-chain deprotection, and peptide work-up [3,4]. Based on previous laboratory studies on PPGs [5], we decided to use them as the third-dimensional protecting groups for amino acid side chains to verify the possibility of their application in cyclic peptide synthesis.

PPGs enjoyed a resurgence in interest after the publications of Kaplan [6] and Engels [7] in the late 1970s [8]. PPGs have an interesting feature: they do not require any reagent for their cleavage, only UV light. A fundamental role is played by PPGs in the process of greening. This category of protecting groups opens the possibility of dealing with sensitive molecules that are otherwise incompatible with acids or bases [9]. Moreover, the greening use of PPGs in nucleic acid [10], carbohydrate [11], and peptide [12,13,14,15] chemistry has been well-established [16]. In contrast to chemically cleavable protecting groups, PPGs use light as a traceless tool in the deprotection process [17], which not only simplifies the reaction step but also improves the purity and operability of the final product. Simultaneously, as the third-dimensional protective group, PPGs are convenient tools for the site-specific modification of peptides and the preparation of cyclic peptides, which also play significant roles in the structural modification of drugs. The most commonly used photolabile groups are currently *o*-nitrobenzyl derivatives, which have proven to be highly versatile and are used to protect a wide variety of functional groups [8,18]. Pfleiderer et al. [16] synthesized a novel type of PPG, 2-(*o*-nitrophenyl)-propyloxycarbonyl [19,20] (Nppoc), which was used to protect the hydroxyl groups in nucleosides and to conduct a series of nucleoside modifications. Unlike the conventional *o*-nitrobenzyl structural group, the photocyclization of Nppoc [19] (Figure 1) to N-hydroxyindole ketone has been postulated to involve the initial formation of nitrostyrene products that could cyclize with the cleavage of the C_β_-O bond [21] because several of the side products are completely inactive, which is advantageous in the synthesis of cyclic peptides. Additionally, Steiner et al. [21] reported that, when the benzylic site was alkylated, the photolysis quantum yield was significantly higher.

Recently, we used the 2-(*o*-nitrophenyl) propan-1-ol (Npp-OH) group—a rapid and effective PPG for carboxyl groups in standard solid-phase peptide synthesis (SPPS) that is tolerated in the reaction conditions used to cleave the *t*-Bu ester and the Fmoc group and is rapidly removed by UV irradiation without any unwanted side reactions in peptide synthesis [5]. However, because Npp-OH is an aliphatic primary alcohol, its homologous ester is easily attacked by nucleophiles. Additionally, the diketopiperazine side reaction (DKP) [22] is already present in Npp-OH in some situations, which has limited further applications.

In order to solve the abovementioned problems, we modified the benzyl site of Npp-OH to transform it into a secondary alcohol and obtained a series of derivatives. Furthermore, photosensitivity, acid–base tolerance, and applicability tests of the Npp-OH derivatives (compounds **4** [19], **5**, **12** [19], **13**, **16**, and **17**) as carboxyl-protective groups were carried out. Finally, the fast and stable Npb-OH photolabile protecting group was selected for the third-dimensional protective group of the amino acid side chain and successfully applied in cyclic peptide synthesis. It was also verified that, in addition to playing an important role in the structural modification of drugs, the photoprotective group is a convenient tool for the site-specific modification of polypeptides and the preparation of cyclic peptides.

## 2. Results and Discussion

### 2.1. Synthesis of the Photolabile Protecting Groups (PPGs)

Following previous work based on the Npp-OH photolabile group, six modified derivatives (see **4** [19], **12** [19], **13**, **16**, **17**, and **5**) substituted at Npp-OH side chain were synthesized. The derivatives involved the preparation of a series of intermediates and products, ten of them were known compounds including **1** [23], **2** [19], **3** [24], **4**, **7** [25], **9** [26], **10** [19], **12** [19], **14** [27] and **15** [27]. The ten compounds were synthesized in different methods from the corresponding literature. For example, the potassium carbonate in the literature was replaced with the sodium hydroxide for the synthesis of 1, and the yield remains at 84.7%. **4** was obtained by reduction of **2** in Figure 2, and the preparation of **2** used NaH instead of sodium metal, making the reaction conditions milder. Compound **12** was obtained from the reduction of **10** in Figure 3, and the preparation of **10** used potassium tert-butoxide instead of sodium metal to make the reaction conditions milder. In the literature, **3** was synthesized through two steps with 2-nitrophenyl acetic acid as the material. While in this paper, **3** was synthesized through one step with 1 as the material, and the reaction temperature was reduced from 100 ℃ to 60 ℃. In the synthesis of **7**, we tried to replace the potassium hydroxide with potassium tert-butanol, and the yield was slightly reduced. In the synthesis of **9**, the starting materials and synthesis route were different from the literature, and the yield was increased from 66% to 79.4%. In the synthesis of **14** and **15**, the starting materials and the synthesis route were different from the literature methods with the relatively mild reaction condition of 75 ℃ in tetrahydrofuran replacing the condition of −40 ℃ under nitrogen. The following were the corresponding synthetic routes. The synthesis of PPGs **4** [19] and 5 is illustrated in Figure 2. Under alkaline conditions, both compounds were subjected to nucleophilic substitution reactions using ethyl acetoacetate and 2-fluoronitrobenzene as starting materials and then hydrolyzed to obtain **1**, which was alkylated or alkylidene in the benzylic position with MeI or polyoxymethylene to form the corresponding ketones. The NaBH_4_ or NaBH4/CeCl_3_·7H_2_O reduction of **2** or **3** produced benzylic 1-monosubstituted and 1,2-disubstituted 1-(2-nitrophenyl) propan-2-ol. When CeCl_3_·7H_2_O was used as the reducing agent, only the ketone was reduced, which had no effect on the double bond [28]. The structures of the products were assigned with ^1^H NMR, ^13^C NMR, and HRMS. Even though compounds **12** and **13** were also modified in the benzylic position, they have different synthetic routes in Figure 3. We first tried to use 2-nitroethylbenzene as a raw material but found that the addition reaction for the ethyl group in 2-nitroethylbenzene did not proceed normally. Therefore, under base conditions, after inducing a carbon anion addition reaction, we reacted 2-nitrotoluene with benzaldehyde and pivalaldehyde to obtain **6** and **7**, which were then converted to the corresponding ketones **8** and **9** in high yields via oxidation with PCC. Subsequently, using CH_3_I as a raw material, an alkylation reaction was induced at the exocyclic 1-position to obtain **10** and **11**. The NaBH_4_ reduction of **10** and **11** led to high yields of exocyclic 1-monosubstituted 2-(2-nitrophenyl) propanol **12** and **13**. Previously, we tried to use the synthesis method using Npp-OH, benzaldehyde, and valeral instead of paraformaldehyde to participate in the reaction and prepare Nppp-OH (**12**) and Dmnp-OH (**13**), but we found that the reaction could not be carried out normally. Even when the alkali was replaced with sodium glycolate, potassium tert-butanol, and potassium hydroxide-18-crown-6, the reaction still failed. When sodium glycolate was used as a base, carbanions could not form. Moreover, the use of potassium tert-butanol and potassium hydroxide-18-crown-6 allowed the carbanion to form smoothly, but the subsequent addition reaction could not continue, which was probably caused by the steric hindrance effect of ethyl and aldehyde substituents (Figure 4). However, we replaced the reaction substrates with 2-nitrotoluene (which presented a deep red color under the influence of alkali) and showed that carbon anions formed; then, benzaldehyde and valeraldehyde were added, and the reaction could occur smoothly. Thus, the failure of the one-step synthesis of **12** and **13** was due to the ethyl in 2-nitroethylbenzene, whose steric-hindrance effect led to the failure of addition.

Using cyclones and tetrahydropyrrole as substrates, enamine products were obtained via the alkylation of enamine and purification by vacuum distillation; these products were mixed with 2-fluoronitrobenzene to obtain compounds **14** and **15** via an acylation reaction. A formal exocyclic 1,2-disubstitution was also observed in 4-(2-hydroxypentyl)- and 4-(2-hydroxyhexyl)-substituted 1-dinitrobenzenes **16** and **17**, which resulted from **14** and **15** via NaBH_4_ reduction in the form of the compound shown in Figure 5. 

### 2.2. Characterization of PPGs 

#### 2.2.1. Acid and Alkali Stability of the PPGs

To study the acid–base sensitivity of the newly synthesized *o*-nitrobenyl-alcohols as multitalented PPGs in peptide chemistry, all the *o*-nitrobenyl alcohols were acetylated to form the corresponding esters (Figure 6). Then, all the esters were dissolved in a TFA solution and 20% piperidine–DMF for 24 h under ambient temperature and monitored via TLC. As a result, considering the yield of the esterification and acid–base stability, **4a** and **5a** exhibited high yields, and **12a**, **13a**, **16a**, and **17a** were stable in an alkaline solution and unstable in an acidic solution. These results indicate that the active side chain containing **4a** and **5a** could be exposed for Fmoc-SPSS under acidolysis-inducing conditions and in piperidine solution (Table 1). Therefore, compounds **4a** and **5a** are protective groups suitable for peptide synthesis.

#### 2.2.2. Photolysis of Esterified Compounds (**4**) and (**5**)

Afterwards, to research the photolability of **4** and **5**, we utilized conventional conditions for photolysis experiments. Two *o*-nitrobenyl-alcohols were separately condensed with Fmoc-Ala-OH to form the corresponding Fmoc-Ala-COPPG esters (Figure 7), which were dissolved in methanol and irradiated at 365 nm for 30 min using a UV lamp (6 W); the reactions were quantitatively monitored with TLC. 

We observed that **5** could not be released, whereas **4** could be completely released, which suggested that **4** has high photolytic efficiency. A possible mechanism of photolysis (Figure 8), based on that of Npp [16,21], was inferred here. Under the photolysis conditions, compound **4** first underwent hydrogen transfer at the α site, and subsequent β-elimination led to the expected o-nitrostyrene product and the release of the alcohol (ROH) from the carbonate. Of course, compound **5** could not undergo hydrogen transfer due to the absence of hydrogen at the α site. Therefore, compound **4** had a high photolysis rate, and compound **5** had a poor photolysis rate. The results also demonstrate that the presence of α-H at the *o*-nitrophenyl α site was necessary for the photolability of the *o*-nitrophenyl propanol compounds. 

Next, we used RP-HPLC to further investigate the photochemical properties during irradiation at 365 nm. Here, a 0.2 mmol solution of **4a** and **5a** in DMF–MeOH (3:1, *v*/*v*) was prepared to study the progress of photolysis, which was carried out with a LED-UV (365 nm and 350 W) for 0, 5, 10, and 20 min. The analysis was performed on a Venusil ASB C_18_ column (5 × 150 × 4.6 mm) with CH_3_CN-H_2_O (0.1% TFA) as the eluent system (10–60%, 20 min). The photolysis process is shown in Figure 1. During irradiation, the **4a** peak gradually disappeared as the reaction proceeded, and it was substituted by a new peak corresponding to the photolysis product (retention time: 25.043 min). After 20 min, **4a** was completely photolyzed, but **5a** failed to complete photolysis. These results indicate that the photolysis reaction had no side reactions and that **4** had a better photolytic efficiency than **5**. 

#### 2.2.3. DKP Side Reactions

One of the most encountered side reactions in any strategy used for Wang resins is the formation of diketopiperazine (DKP) (2,5-piperazinediones) [29,30,31,32]. The configuration, conformation, and steric hindrance of amino acids all play important roles in SPPS, as DKP formation is especially prone to occur when the solid-phase carrier is an ester junction arm. Especially when the first amino acid is a sterically unhindered amino acid (e.g., glycine), the free NH_2_ of the second amino acid (e.g., proline) with a cis-amide conformation can easily attack the ester structure, leading to the premature removal of the peptide. Its stable six-membered ring provides the thermodynamic force of the reaction. When we fixed the side chain of amino acids to prepare cyclic peptides, this reaction led to the dissociation of the carboxy-terminal protective group. The side effect resulted in both a lower yield and the crude reaction of several deletion peptides lacking the first amino acids. Thus, we designed the Glu–Pro dipeptide as a template compound and used Npb-ester as a side-chain-protection strategy to avoid the DKP side reaction. We attempted to protect the first Fmoc-glutamic acid side-chain carboxyl group supported on Wang resin with Npb-OH (**4**) and Npp-OH, then loaded it with Fmoc-Pro-OH, and finally deprotected it with 20% piperidine–DMF. Via TLC monitoring, we found that the Npp-OH PPG was released, but **4** was not liberated. We believe that **4** can effectively counteract the occurrence of DKP side reactions because, in our study, when the C-terminal amino acid was linked via a non-bulky ester group (e.g., Wang resin), the formation of DKP was often accompanied by the deprotection of the second amino acid residue and subsequent acylation, resulting in the release of PPGs (Figure 9). Therefore, compound **4** is an appropriate PPG for peptide synthesis.

#### 2.2.4. Applicability of PPG **4**

We next investigated compound **4′**s addition to carboxyl acids with a wide range of esterification protocols. Based on previous studies, the ester was prepared with the DCC–DMAP method (Table 2). The esterification of carboxyl groups by **4** enabled the corresponding esters to be obtained in higher yields and esters to be stable during the acid removal of the side chains, together demonstrating compound **4** to be a carboxyl-protecting group with wide adaptability. Therefore, the Npb-OH photolabile protecting group was considered to be a third-dimensional protective group for Glu residue side-chain protection to be applied in cyclic peptide synthesis.

### 2.3. Application of PPGs in Cyclic Peptide Synthesis

Anti-inflammatory peptides **I** [33], **II** [34], and **III** [35], all containing Glu residues, were chosen to illustrate the use of the Npb-OH strategy in head-to-tail cyclic peptide **I** and head-to-side cyclic peptide **II** and **III** synthesis (Figure 10 and Table 3). Firstly, we photo-protected the carboxyl group of the Fmoc-glutamic acid side chain with **4** via an esterification reaction, and then, we removed the *t*-butyl side chain (which was assembled on the peptide chain via Fmoc SPPS) under standard acidic conditions. The deprotection of the N-terminus of Fmoc was conducted using 20% piperidine–DMF, and the formation of the peptide bond was accomplished with the Fmoc-SPSS strategy. After the completion of peptide assembly, the photo-deprotection reaction was carried out with a LED-UV (365 nm and 350 W) twice for 15 min before the carboxyl group was revealed. The solvent was DMF: MeOH (3:1, *v*/*v*), which was degassed under an Ar atmosphere. When we used DMF as a solvent to photolyze the peptide, we needed to add other low-boiling solvents to counter the thermal effects of the UV lamp, so before photolysis, argon gas was pre-bubbled into the solvent to deoxidize. At the same time, 15 min after the first photolysis, the product released via photolysis caused the solution to change from colorless to yellow. The solution was filtered out and replaced with the new solvent, and the second photolysis was carried out for 15 min. The solution remained colorless, indicating that the deprotection reaction had reached the endpoint. Next, the exposed carboxyl group and the N-terminal amino group formed a lactam ring on the resin via a condensation reaction. The cyclic peptide was split from the resin (Figure 10). The lysate was concentrated, ice diethyl ether was added, and the solid was collected by centrifugation and blow-dried with argon. The purity and structure were confirmed using analytical HPLC (see Appendix A, Appendix A) and Q-FT-ICR–MS (Table 3).

## 3. Materials and Methods

Room temperature (r.t) refers to ambient temperature. Solid-phase synthesis was manually carried out in a polypropylene syringe containing a polyethylene frit. The visualization of the TLC was performed using UV light. The photolysis was performed via a LED-UV (365 nm and 350 W). Flash column chromatography was performed on silica gel (200–300 mesh). The ^1^H NMR and ^13^C NMR spectra were recorded with a Bruker AM 400 and AM 500 MHz spectrometer (Palo Alto, CA, USA). The chemical shifts (δ) are reported in ppm, with an internal standard of tetramethylsilane (0 ppm). The coupling constants (J) are provided in hertz. Multiplets were designated as s (singlet), d (doublet), t (triplet), q (quartet), and m (multiplet). High-resolution mass spectrometry (HRMS) measurements were recorded using an Agilent 1260-G6230A ESI–HQMS spectrometer (Agilent, USA). Analytical high-pressure liquid chromatography (HPLC) was carried out with an Agilent instrument (Agilent1100, Agilent, USA), automatic injector, photodiode array detector, system controller (Empower login, Agilent, USA), and Venusil ASB C18 column (5 mm, 150 × 4.6 mm, Bonna-Agela Technologies, Tianjin, China). UV measurements were recorded at 220 nm.

### 3.1. Synthesis of the Photolabile Protecting Groups (PPGs)

#### General Reduction Procedure

NaBH_4_ (0.40 g, 10.6 mmoL) was slowly added to a suspension of 3-(2-nitrophenyl) butan-2-one **2** (1.02 g, 5.3 mmoL) in MeOH (10 mL) while the temperature was kept below 0 °C for 30 min. After stirring for 3 h at r.t, the mixture was acidified with a HCl solution (2 M, 10 mL), extracted with DCM (3 × 10 mL), and washed with saturated brine. The combined organic layer was dried over MgSO_4_ and evaporated, and the crude product was purified with flash column chromatography (column chromatography eluent; petroleum ether:EtOAc = 15:1) to yield product **4** [19]. The new compounds **12 [19]**, **13**, **16**, **17**, and **5** were characterized as follows.

*3-(o-Nitrophenyl)butan-2-ol (***4***)*: 1.54 g, 80.2% yield. Column chromatography eluent, PE:EA = 10:1. Red-brown oil. ^1^H-NMR (400 MHz, DMSO-*d*_6_) δ 7.75 (d, *J* = 8.4 Hz, 1H), 7.67–7.60 (m, 2H), 7.36 (t, *J* = 6.9 Hz, 1H), 4.61 (d, *J* = 4.9 Hz, 1H), 3.76–3.70 (m, 1H), 2.99 (p, *J* = 7.1 Hz, 1H), 1.23 (d, *J* = 7.0 Hz, 3H), 1.04 (d, *J* = 6.2 Hz, 3H); ^13^C NMR (150 MHz, DMSO-*d*_6_) δ 151.0, 138.5, 132.2, 129.0, 126.8, 123.1, 69.7, 40.7, 21.6, 17.7; HRMS (ESI-TOF) calculated for C_10_H_13_NO_3_ [M+Na]^+^ 218.0793, Found 218.0788. The structure of product **4** was confirmed by the ^1^H- and ^13^C-NMR spectra (Appendix A).

*3-(o-Nitrophenyl)but-3-en-2-ol (***5***)*: 2.34 g, 68.2% yield. Column chromatography eluent, PE:EA = 10:1. Light brown oil. ^1^H-NMR (400 MHz, DMSO-*d*_6_) δ 7.86 (d,1H,Ar-H), δ 7.63 (t, 1H, Ar-H), δ 7.87 (d, *J* = 8.1 Hz, 1H), 7.64 (t, *J* = 7.6 Hz, 1H), 7.50 (t, *J* = 8.0 Hz, 1H), 7.40 (d, *J* = 7.7 Hz, 1H), 5.35 (d, *J* = 17.3 Hz, 1H), 5.08 (d, *J* = 5.0 Hz, 1H), 4.92 (d, *J* = 17.3 Hz, 1H), 4.40–4.34 (m, 1H), 1.07 (d, *J* = 6.4 Hz, 3H); ^13^C NMR (150 MHz, DMSO-*d*_6_) δ 151.1, 149.3, 135.4, 133.0, 131.7, 129.0, 124.1, 113.6, 68.5, 23.2; HRMS (ESI-TOF) calculated for C_10_H_11_NO_3_ [M+H]^+^ 194.0817, Found 194.0812. The structure of product **5** was confirmed by the ^1^H- and ^13^C-NMR spectra (Appendix A).

*2-(2-nitrophenyl)-1-phenylpropan-1-ol (***12***)*: 1.59 g, 83.0% yield. Column chromatography eluent, PE:EA = 10:1. Light yellow oil. ^1^H NMR (400 MHz, DMSO-*d*_6_) δ 7.74 (dd, *J* = 12.0, 8.1 Hz, 2H), 7.66 (t, *J* = 7.6 Hz, 1H), 7.42 (t, *J* = 7.6 Hz, 1H), 7.32–7.22 (m, 5H), 5.32 (s, 1H), 4.59 (d, *J* = 8.5 Hz, 1H), 3.49–3.42 (m, 1H), 1.05 (d, *J* = 7.0 Hz, 3H); ^13^C NMR (150 MHz, DMSO-*d*_6_) δ 151.5, 144.8, 138.8, 132.8, 129.3, 128.4, 128.4, 127.7, 127.4, 127.1, 127.1, 123.7, 77.3, 41.2, 18.3; HRMS (ESI-TOF) calculated for C_15_H_15_NO_3_ [M-H]^−^ 256.0974, Found 256.0967. The structure of product **12** was confirmed by the ^1^H- and ^13^C-NMR spectra (Appendix A).

*2,2-dimethyl-4-(2-nitrophenyl)pentan-3-ol (***13***)*: 1.82 g, 85.4% yield. Column chromatography eluent, PE:EA = 6:1. Yellow oil. ^1^H NMR (400 MHz, DMSO-*d*_6_) δ 7.85 (dd, 1H), δ 7.60 (dd, 1H), δ 7.49 (m, 1H), δ 7.43 (m, 1H), δ 4.56 (d, 1H), δ 3.14 (m, 1H), δ 3.09 (d, 1H), δ 2.72 (dd, 1H), δ 0.88 (s, 9H); ^13^C NMR (150 MHz, DMSO-*d*_6_) δ 150.9, 138.9, 132.0, 131.8, 127.3, 123.3, 81.8, 36.1, 34.3, 27.2, 27.2 27.2, 23.2; HRMS (ESI-TOF) calculated for C_13_H_19_NO_3_ [M-H]^−^ 236.1285, Found: 236.1289. The structure of product **13** was confirmed by the ^1^H- and ^13^C-NMR spectra (Appendix A).

*2-(o-Nitrophenyl)cyclopentan-1-ol (***16***)*: 0.48 g, 46.2% yield. Column chromatography eluent, PE:EA = 10:1. Yellow oil. ^1^H-NMR (400 MHz, DMSO-*d*_6_) δ 7.77 (d, *J* = 8.0 Hz, 1H), 7.67–7.59 (m, 2H), 7.43 (t, *J* = 7.0 Hz, 1H), 4.82 (s, 1H), 4.11 (q, *J* = 7.1 Hz, 1H), 3.11 (dt, *J* = 10.1, 7.9 Hz, 1H), 2.16–2.08 (m, 1H), 2.00–1.92 (m, 1H), 1.79–1.69 (m, 2H), 1.60–1.49 (m, 2H); ^13^C NMR (150 MHz, DMSO-*d*_6_) δ 151.4, 138.2, 133.17, 128.9, 127.5, 123.8, 78.6, 48.5, 34.7, 32.7, 22.2; HRMS (ESI-TOF) calculated for C_11_H_13_NO_3_ [M+Na]^+^ 230.0793, Found 230.0785. The structure of product **16** was confirmed by the ^1^H- and ^13^C-NMR spectra (Appendix A).

*2-(o-Nitrophenyl)cyclohexan-1-ol (***17***)*: 1.74 g, 60.8% yield. Column chromatography eluent, PE:EA = 10:1. Yellow oil. ^1^H-NMR (400 MHz, DMSO-*d*_6_) δ 7.74 (d, *J* = 7.8 Hz, 1H), 7.63 (d, *J* = 8.0 Hz, 2H), 7.40 (dq, *J* = 8.4, 4.2 Hz, 1H), 4.52 (d, *J* = 5.6 Hz, 1H), 3.59–3.53 (m, 1H), 2.78 (td, *J* = 12.1, 10.0 Hz, 1H), 1.93 (dd, *J* = 11.8, 4.3 Hz, 1H), 1.84 (dd, *J* = 12.5, 3.9 Hz, 1H), 1.74–1.67 (m, 2H), 1.54–1.31 (m, 2H), 1.28–1.17 (m, 2H); ^13^C NMR (150 MHz, DMSO-*d*_6_) δ 151.6, 139.0, 132.9, 128.9, 127.2, 123.8, 72.3, 46.7, 36.6, 33.2, 26.1, 25.0; HRMS (ESI-TOF) calculated for C_12_H_15_NO_3_ [M+Na]^+^ 244.0950, Found 244.0942. The structure of product **17** was confirmed by the ^1^H- and ^13^C-NMR spectra (Appendix A).

### 3.2. Photolysis of Esterified Compounds (**4**) and (**5**)

In a 50 mL round-bottomed quartz bottle, 30 mL of 0.2 mmol **4a** and **5a** in DMF-MeOH (3:1, *v*/*v*) solution were added to study the photolysis process, respectively, which was carried out with a LED-UV (365 nm and 350 W) for 0, 5, 10, and 20 min. The photolysis experiment used LED-UV (365 nm, 700 W), and the actual photolysis power was 50% of the rated power, which was 350 W. The analysis was performed on a Venusil ASB C18 column (5 × 150 × 4.6 mm) with CH_3_CN-H_2_O (0.1% TFA) as the eluent system (10–60%, 20 min).

### 3.3. Applicability of PPG **4**

#### 3.3.1. General Esterification Procedure

A solution of carboxylic acid (5 mmol), DMAP (0.5 mmol), and Npb-OH (5 mmol) in 50 mL of DCM was cooled while stirring in an ice bath. DCC (6 mmol) was added in small portions, and the reaction mixture was stirred at 0 °C for 30 min and then at r.t for 2 h. The solution was filtered, the filtrate was concentrated to dryness under vacuum, and the residue was purified with flash column chromatography. The new compounds **4b**, **4e**, **4f**, **4g**, and **4h** were characterized as follows.

*3-(2-nitrophenyl)butan-2-yl(((9H-fluoren-9-yl)methoxy)carbonyl)alaninate (***4b***)*. 1.30 g, 83.4% yield. Column chromatography eluent, PE:EA = 6:1. Light yellow oil. ^1^H NMR (400 MHz, DMSO-*d*_6_) δ 7.90 (d, *J* = 7.5 Hz, 2H), 7.79 (d, *J* = 7.5 Hz, 1H), 7.71–7.67 (m, 3H), 7.65–7.60 (m, 2H), 7.46–7.40 (m, 3H), 7.35–7.28 (m, 2H), 5.06 (m, 1H), 4.29–4.17 (m, 3H), 3.82 (q, *J* = 7.3 Hz, 1H), 3.24 (q, *J* = 7.3 Hz, 1H), 1.29 (d, *J* = 5.6 Hz, 3H), 1.19 (d, *J* = 5.5 Hz, 3H), 1.17 (dd, *J* = 8.1, 5.5 Hz, 3H); ^13^C NMR (150 MHz, DMSO-*d*_6_) δ 172.6, 156.2, 151.1, 144.2, 143.0, 141.2, 141.2, 137.1, 133.0, 129.4, 129.0, 129.0, 128.9, 128.9, 128.1, 127.5, 127.5, 125.7, 120.6, 120.6, 74.6, 60.2, 50.1, 47.0, 39.2, 18.7, 18.1, 16.5; HRMS (ESI-TOF) calculated for C_28_H_28_N_2_O_6_ [M+H]^+^ 489.2026, Found 489.2019. The structure of **4b** was confirmed by the ^1^H- and ^13^C-NMR spectra (Appendix A).

*3-(2-nitrophenyl)butan-2-yl(((9H-fluoren-9-yl)methoxy)carbonyl)phenylalaninate (***4e***)*. 2.28 g, 80.3% yield. Column chromatography eluent, PE: EA = 6:1. Light brown oil. ^1^H NMR (400 MHz, DMSO-*d*_6_) δ 7.88 (d, *J* = 7.6 Hz, 2H), 7.82–7.77 (m, 2H), 7.70 (d, *J* = 8.1 Hz, 1H), 7.66–7.61 (m, 2H), 7.60 (d, *J* = 7.6 Hz, 1H), 7.45–7.38 (m, 3H), 7.31–7.24 (m, 4H), 7.21–7.11 (m, 3H), 5.11 (dq, *J* = 8.7, 6.0 Hz, 1H), 4.19–4.11 (m, 3H), 3.99 (dt, *J* = 12.2, 7.9 Hz, 1H), 3.31–3.24 (m, 1H), 2.37 (dd, *J* = 13.8, 11.0 Hz, 1H), 2.24 (dd, *J* = 13.7, 3.9 Hz, 1H), 1.24 (d, *J* = 7.0 Hz, 3H), 1.18 (d, *J* = 6.2 Hz, 3H); ^13^C NMR (150 MHz, DMSO-*d*_6_) δ 171.7, 156.3, 151.2, 144.1, 144.1, 141.1, 141.1, 137.9, 137.0, 133.1, 129.4, 129.4, 129.1, 128.7, 128.7, 128.7, 128.1, 128.1, 127.5, 127.5, 127.0, 125.7, 120.6, 120.6, 75.2, 66.1, 56.1, 47.0, 39.0, 36.0, 18.7, 18.0; HRMS (ESI-TOF) calculated for C_34_H_32_N_2_O_6_ [M+H]^+^ 565.2339, Found 565.2333. The structure of **4e** was confirmed by the ^1^H- and ^13^C-NMR spectra (Appendix A).

*3-(2-nitrophenyl)butan-2-yl 2-(2-nitrophenyl)acetate (***4f***)*. 1.68 g, 93.6% yield. Column chromatography eluent, PE:EA = 6:1. Light yellow/brown oil. ^1^H NMR (400 MHz, DMSO-*d*_6_) δ 8.08 (d, *J* = 7.8Hz, 1H), 7.73 (dd, *J* = 7.8, 1.4 Hz, 2H), 7.60–7.51 (m, 2H), 7.45 (d, *J* = 1.4 Hz, 2H), 7.39 (d, *J* = 7.8 Hz, 1H), 5.03–5.00 (m, 1H), 3.89 (d, *J* = 7.1 Hz, 2H), 3.26 (t, *J* = 7.2 Hz, 1H), 1.23 (d, *J* = 7.0 Hz, 3H), 1.08 (d, *J* = 6.2 Hz, 3H); ^13^C NMR (150 MHz, DMSO-*d*_6_) δ 169.3, 151.0, 148.5, 136.5, 134.5, 134.1, 132.9, 130.2, 129.3, 129.2, 128.1, 125.3, 123.8, 74.6, 39.4, 38.3, 18.4, 18.0; HRMS (ESI-TOF) calculated for C_18_H_18_N_2_O_6_ [M+H2O]^+^ 376.1271, Found: 376.1504. The structure of **4f** was confirmed by the ^1^H- and ^13^C-NMR spectra (Appendix A).

*3-(2-nitrophenyl)butan-2-yl nicotinate (***4g***)*. 1.35 g, 89.6% yield. Column chromatography eluent, PE:EA = 6:1. Light brown oil. ^1^H NMR (400 MHz, DMSO-*d*_6_) δ 8.85 (d, *J* = 2.2 Hz, 1H), 8.77 (dd, *J* = 4.8, 1.7 Hz, 1H), 8.07 (dt, *J* = 8.0, 2.0 Hz, 1H), 7.78 (dd, *J* = 8.1, 3.3 Hz, 2H), 7.65 (t, *J* = 4.8 Hz, 1H), 7.51 (dd, *J* = 4.8, 1.7 Hz, 1H), 7.41 (t, *J* = 4.8 Hz, 1H), 5.36 (tq, *J* = 8.8, 6.2 Hz, 1H), 3.49–3.41 (m, 1H), 1.39 (d, *J* = 5.4 Hz, 3H), 1.35 (d, *J* = 5.4 Hz,3H); ^13^C NMR (150 MHz, DMSO-*d*_6_) δ 164.0, 151.0, 150.4, 147.8, 137.8, 136.9, 133.1, 129.5, 128.2, 127.8, 124.9, 123.8, 75.3, 38.9, 18.6, 18.2; HRMS (ESI-TOF) calculated for C_16_H_16_N_2_O_4_ [M+H]^+^ 301.1188, Found 301.1185. The structure of **4g** was confirmed by the ^1^H- and ^13^C-NMR spectra (Appendix A).

*3-(2-nitrophenyl)butan-2-yl picolinate (***4h***)*. 1.40 g, 93.3% yield. Column chromatography eluent, PE:EA = 6:1. Light yellow oil. 1H NMR (400 MHz, DMSO-*d*_6_) δ 8.69 (d, *J* = 4.8 Hz, 1H), 7.94 (t, *J* = 7.7 Hz, 1H), 7.85 (d, *J* = 7.9 Hz, 1H), 7.77 (t, *J* = 4.8 Hz, 2H), 7.67–7.58 (m, 2H), 7.42 (t, *J* = 7.8 Hz, 1H), 5.37 (p, *J* = 6.3 Hz, 1H), 3.48 (q, *J* = 7.2 Hz, 1H), 1.37 (d, *J* = 8.9 Hz, 3H), 1.31 (d, *J* = 6.3 Hz, 3H); 13C NMR (150 MHz, DMSO-*d*_6_) δ 164.2, 154.2, 151.1, 150.2, 137.0, 136.0, 133.1, 129.0, 128.2, 125.8, 124.3, 123.7, 75.8, 39.1, 18.7, 17.9; HRMS (ESI-TOF) calculated for C16H16N2O4 [M+H]^+^ 301.1188, Found 301.1183. The structure of **4h** was confirmed by the ^1^H- and ^13^C-NMR spectra (Appendix A).

#### 3.3.2. General TFA Deprotection Procedure

The tBu-protected Npb-ester (5 mmol) was dissolved in TFA and stirred until the reaction was completed (1 h). The solvent was removed under reduced pressure. The crude residue was purified with flash column chromatography to yield the product as a yellow oil or an amorphous yellow solid. The new compounds **4c** and **4d** were characterized as follows.

*3-(2-nitrophenyl)butan-2-yl(((9H-fluoren-9-yl)methoxy)carbonyl)serinate (***4c***)*. 1.82 g, 72.1% yield. Column chromatography eluent, PE:EA = 2:1. Yellow oil. ^1^H NMR (400 MHz, DMSO-*d*_6_) δ 7.90 (d, *J* = 7.5 Hz, 2H), 7.79–7.74 (m, 1H), 7.71 (d, *J* = 7.5 Hz, 2H), 7.66–7.61 (m, 2H), 7.53–7.50 (m, 1H), 7.46–7.39 (m, 3H), 7.33–7.30 (m, 2H), 5.11–5.02 (m, 1H), 4.28–4.19 (m, 3H), 3.98–3.95 (m, 1H), 3.55 (d, *J* = 7.5 Hz, 1H), 3.35–3.22 (m, 3H), 1.27 (d, *J* = 6.2 Hz, 3H), 1.10 (dd, *J* = 9.3, 5.6 Hz, 3H); ^13^C NMR (150 MHz, DMSO-*d*_6_) δ 170.3, 156.5, 151.0, 144.3, 144.3, 141.2, 141.2, 136.6, 133.0, 128.1, 128.1, 128.1, 127.5, 127.5, 127.5, 127.5, 125.7, 125.7, 120.6, 120.6, 74.4, 66.2, 61.1, 57.4, 47.0, 38.6, 18.4, 18.0; HRMS (ESI-TOF) calculated for C_28_H_28_N_2_O_7_ [M+Na]^+^ 527.1794, Found 527.1792. The structure of **4c** was confirmed by the ^1^H- and ^13^C-NMR spectra (Appendix A).

*2-((((9H-fluoren-9-yl)methoxy)carbonyl)amino)-5-((3-(2-nitrophenyl)butan-2-yl)oxy)-5-oxopentanoic acid (***4d***)*. 2.48 g, 90.8% yield. Column chromatography eluent, PE:EA = 2:1. Light yellow oil. ^1^H NMR (400 MHz, DMSO-*d*_6_) δ 12.63 (s, 1H), 7.90 (d, *J* = 7.6 Hz, 2H), 7.75–7.68 (m, 3H), 7.66–7.58 (m, 3H), 7.44–7.38 (m, 3H), 7.29 (t, *J* = 7.5 Hz, 2H), 5.03–4.98 (m, 1H), 4.31–4.20 (m, 3H), 3.90–3.85 (m, 1H), 3.32–3.29 (m, 1H), 2.10–2.21 (m, 2H), 1.82–1.79 (m, 1H), 1.66–1.60 (m, 1H), 1.30 (d, *J* = 7.0 Hz, 3H), 1.19 (d, *J* = 6.9 Hz, 3H); ^13^C NMR (150 MHz, DMSO-*d*_6_) δ 173.8, 171.1, 156.4, 151.1, 144.3, 144.2, 141.18, 136.8, 136.3, 132.3, 129.6, 129.2, 128.1, 128.1, 127.5, 127.5, 125.7, 125.7, 123.8, 120.6, 120.6, 74.7, 66.1, 60.2, 47.1, 38.9, 31.2, 21.2, 17.7, 14.6; HRMS (ESI-TOF) calculated for C_30_H_30_N_2_O_8_ [M-H]^−^ 545.1924, Found 545.1927. The structure of **4d** was confirmed by the ^1^H- and ^13^C-NMR spectra (Appendix A).

### 3.4. Application of PPGs in Cyclic Peptide Synthesis

#### 3.4.1. General Procedure for Solid-Phase Reactions

The peptide was synthesized by using 2.005 g of resin (0.11 mmol) as a starting material, followed by (a) Fmoc deprotection with 20% piperidine–DMF (6 mL, 3 equiv.) for 15 min and washing in DMF (6 mL × 3 × 30 s), and (b) establishing peptide-coupling conditions comprising three equivalents of HOBt, three equivalents of TBTU, three equivalents of Fmoc-amino acid, and three equivalents of NMM in DMF (6 mL) for 2 h at r.t, followed by washing in DMF (6 mL × 3 × 30 s). The coupling reaction was monitored with the Kaiser test.

#### 3.4.2. Removal of the Npb Group via Photolysis Experiment

After the last amino acid with the Fmoc group was loaded, and the Fmoc group was removed, the resin-bound linear peptide was transferred to a quartz bottle and suspended in 50 mL of DMF–MeOH (3:1, *v*/*v*, 30 mL). The solvent was degassed for 10 min under argon gas to remove the dissolved oxygen. The mixture was then irradiated with a high-pressure Hg lamp (λ _max_ = 365 nm and 250 W) with constant argon bubbling. A reflux condenser was essential because of the heat effect during UV irradiation. After 15 min, the lamp was turned off, and the dark yellow solvent was replaced with fresh solvent mixtures of DMF–MeOH (3:1, *v*/*v*, 30 mL), followed by continuous irradiation until the solvent stopped turning yellow. This procedure usually needed 30 min in total. Finally, the photolyzed resin was collected with filtration and then washed with DCM (3 mL × 3 × 30 s) and DMF (3 mL × 3 × 30 s). A synthesis procedure was performed after irradiating for 15 min, as described next.

#### 3.4.3. Cyclization of Peptides **I**, **II**, and **III**

The resin-bound linear peptide was transferred into a reaction vessel and suspended in 6 mL of DMF; then, HATU (3 equiv.), HOAt (3 equiv.), and NMM (3 equiv.) were sequentially added, and the vessel was capped and shaken at r.t until the Kaiser test was negative. Next, the solvent was removed by filtration, and the resin was washed with DMF (6 mL × 3 × 30 s) and MeOH (6 mL × 3 × 30 s). Heterogeneous 5 mL mixtures of TFA/TIS/H_2_O/ETD/m-Cresol (90:2.5:2.5:2.5:2.5, *v*/*v*/*v*/*v*/*v*) solutions were prepared and added to the reaction vessel. The resulting solution was concentrated under vacuum, and a white precipitate appeared soon after the iced ether was added. The cyclic peptides were analyzed using HPLC at 220 nm with CH_3_CN-H_2_O (0.1% TFA) as the eluent system (Appendix A and Table 3) and identified with ESI–TOF–MS.

## 4. Conclusions

In summary, we synthesized six candidate protective groups with acid–base tolerance, photolability, suitability, and diketopiperazine side-reaction tests. We found that Npb-OH (**4**) was a photolabile protective group that met our needs in all aspects. In particular, it is more stable than Npp-OH (especially in alkaline conditions) and can avoid the side reaction of diketopiperazine. Compound **4** also has the advantages of a high acid–base tolerance, fast photolysis rate, and wide applicability. The protecting group can be removed without additional reagents, and the photolysis reaction is simple, thorough, rapid, and without side reactions. Moreover, **4** has broad applicability in carboxyl protection and subsequent cyclization with peptide terminal amino groups in situ because it can form a lactam bond after removing protection. Ultimately, compared to Npp-OH, **4** has better chemical tolerance and is more suitable for solid-phase peptide synthesis. Additionally, we can regard the Npb-OH photolabile protective group as a green protective group. Furthermore, we successfully synthesized the anti-inflammatory head-to-tail cyclic peptide **I** and head-to-side cyclic peptides **II** and **III** with the Npb-OH group. The results indicate the feasibility of a cyclic peptide-synthesis strategy based on the Npb-OH PPG, which has the advantages of low cost, simple operation, high yield, and purity. The process was successfully applied to the synthesis of cyclic peptides, facilitating a greener process for SPPS. Further structural modifications based on compound **4** to protect -SH and -NH_2_ are ongoing.

## Data Availability

Not applicable.

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
