# Peer review of "Synthesis of Cyclic Peptides in SPPS with Npb-OH Photolabile Protecting Group"

_molecules, 2022, doi:10.3390/molecules27072231_

Round 1
Reviewer 1 Report
Chen et al., were reported a research article entitled as "Synthesis of cyclic peptides in SPSS by using of photolabile protecting group Npb-OH". This research work has novelty and useful method for solid phase peptide synthesis. Although this manuscript was written well, there is a lot of scope to improve representation of the manuscript which is helpful to readers, specifically schemes (Scheme 4 and 5 are not clear and keep consistent spaces, commas).
Author Response
Dear reviewer,Thank you for your valuable comments and questions, and I answer as follows:
Comment 1:Chen et al., were reported a research article entitled as "Synthesis of cyclic peptides in SPSS by using of photolabile protecting group Npb-OH". This research work has novelty and useful method for solid phase peptide synthesis. Although this manuscript was written well, there is a lot of scope to improve representation of the manuscript which is helpful to readers, specifically schemes (Scheme 4 and 5 are not clear and keep consistent spaces, commas).
Response: Thanks for the referee’s kind suggestion. Scheme 4 and 5 have been revised glyphs, commas and then made an overall revision. Yours sincerely,
Lin Wang

Reviewer 2 Report
My opinion is that this paper can be accepted for publication. The results are not exceptional but it is a serious paper and it brings some interesting developments to an already well-known field.
- - in the title SPPS (not SPSS)!
- - line 52: "marvellous" is excessive. May be "interesting".
- - line 57 : UV is not a reagent. Use "agent" or "tool".
- - line 78: ...UV irradiation without any unwanted... (no need for "the occurence of"
- - Figure 2 is unclear. Redraw it with a clear differentiation between 2 and 4 on the one hand, and 3 and 5 on the other. 3 and 5 must appear with a double bond.
- - line 101 : no brackets. Furthermore reaction with H2C=O is not an alkkylation reaction, but an alkylidenation. Rephrase it.
- - line 165: replace "Meanwhile" by "Of course".
- - line 225: ...were choosen to illustrate" (no need for "to be synthesized".
- - general remark: a lot of products described here were previoulsy known. It took me 2 minutes to find a reference about compound 4 (Helvetica Chimica Acta (2004), 87(3), 620-659). This job MUST be done for all molecules and a litterature reference clearly given for all already published compounds. This may be in the experimental part when describing these products. Moreover, it is not clear to me that the spectra of already known compounds should be included in this experimental part.
Author Response
Dear reviewer,Thank you for your valuable comments and questions, and I answer as follows: Comment 1:- in the title SPPS (not SPSS)!
Response: Thanks for the referee’s kind suggestion, the “SPSS” in the title has been changed to “SPPS”.
Comment 2:- line 52: "marvellous" is excessive. May be "interesting".
Response: Thanks for the referee’s kind suggestion, I have changed "marvellous" to "interesting" on line 52.
Comment 3:- line 57 : UV is not a reagent. Use "agent" or "tool".
Response: Thanks for the referee’s kind suggestion, I have changed "reagent" to "tool" on line 57.
Comment 4:- line 78: ...UV irradiation without any unwanted... (no need for "the occurence of"
Response: Thanks for the referee’s kind suggestion, "the occurence of" on line 78 has been removed.
Comment 5:- Figure 2 is unclear. Redraw it with a clear differentiation between 2 and 4 on the one hand, and 3 and 5 on the other. 3 and 5 must appear with a double bond.
Response: Thanks for the referee’s kind suggestion, Figure 2 has been re-edited to clearly show the structure of 2, 3, 4 and 5.
Comment 6:- line 101 : no brackets. Furthermore reaction with H2C=O is not an alkkylation reaction, but an alkylidenation. Rephrase it.
Response: Thanks for the referee’s kind suggestion, the brackets have been removed and the reaction with H2C=O has been revised to an alkylidenation.
Comment7:- line 165: replace "Meanwhile" by "Of course".
Response: Thanks for the referee’s kind suggestion, "Meanwhile" in line 165 has been replaced with "Of course".
Comment 8:- line 225: ...were choosen to illustrate" (no need for "to be synthesized".
Response: Thanks for the referee’s kind suggestion, in line225, "to be synthesized" has been deleted.
Comment 9:- general remark: a lot of products described here were previoulsy known. It took me 2 minutes to find a reference about compound 4 (Helvetica Chimica Acta (2004), 87(3), 620-659). This job MUST be done for all molecules and a litterature reference clearly given for all already published compounds. This may be in the experimental part when describing these products. Moreover, it is not clear to me that the spectra of already known compounds should be included in this experimental part.
Response: Thanks for the referee’s kind suggestion. The references you indicated have been incorporated in the Compound 4 on line 98, and the corresponding references have also been incorporated in the “supplementary materials”. Spectral data for known compounds have also been characterized and attached in the “supplementary materials”.
Reference : 18. Bühler, S.; Lagoja, I.; Giegrich, H.; Stengele, K.P.; Pfleiderer, W. New Types of Very Efficient Photolabile Protecting Groups Based upon the [2-(2-Nitrophenyl)propoxy]carbonyl (NPPOC) Moiety. Helv. Chim. Acta 2004, 87, 620–659, doi:10.1002/hlca.200490060.
Yours sincerely,
Lin Wang

Reviewer 3 Report
Dear Authors,
Your work is excellent and very well written.
Consider the remarks that follow:
Lines 47 & 48: Based on which studies? Please cite respectively. Or maybe consider rephrasing in order to introduce next paragraph.
Line 140: Scheme 6 on Compound 5 & 5a the R1 substitution introduces double equals
Lines 174 & 186: Schemes 8 & 9 are overlapped
Line 216: Adaptive ability should be replaced with adaptability or even better usability?
Finally, it is not directly implicated in your method but I believe providing the optical rotation of compound 4 since it is diastereomer it would add more to your work.
Best regards
Author Response
Dear Reviewer:
Thank you for your valuable comments and questions, and now I will answer as follows:
Comment 1:Lines 47 & 48: Based on which studies? Please cite respectively. Or maybe consider rephrasing in order to introduce next paragraph.
Response: Thanks for the referee’s kind suggestion. Papers on photolabile protecting groups that have been done in this laboratory have been introduced in this section。
Reference : 5. Wang, G.; Peng, T.; Zhang, S.; Wang, J.; Wen, X.; Yan, H.; Hu, L.; Wang, L. 2-(2-Nitrophenyl) propyl: A rapidly released photolabile COOH-protecting group for solid-phase peptide synthesis. RSC Adv. 2015, 5, 28344–28348, doi:10.1039/c5ra01210d.
Comment 2:Line 140: Scheme 6 on Compound 5 & 5a the R1 substitution introduces double equals
Response: Thanks for the referee’s kind suggestion, in Scheme 6, R1 substitutions of compounds 5 and 5a introduced double equals, one has been deleted.
Comment 3:Lines 174 & 186: Schemes 8 & 9 are overlapped
Response: Thanks for the referee’s kind suggestion. The positions of Schemes 8 & 9 have been adjusted to no longer overlap.
Comment 4:Line 216: Adaptive ability should be replaced with adaptability or even better usability?
Response: Thanks for the referee’s kind suggestion. "Adaptive ability" in line 216 has been replaced with "adaptability".
Comment 5:Finally, it is not directly implicated in your method but I believe providing the optical rotation of compound 4 since it is diastereomer it would add more to your work.
Thank you for the referee’s valuable comments. As you pointed out, the focus of this article was to find a photolabile protecting group with better photolysis rate, wide applicability and successful application in cyclic peptide synthesis, so as to to make the cyclic peptide synthesis process greener. At the same time, your opinions are also very important. In the further research and application of Npb-OH, we will characterize the diastereomers of Npb-OH according to your opinions to determine the specific configuration that exerts photosensitivity, making our research more complete and meaningful.
Yours sincerely,
Lin Wang
